# How Individual Variations in the Perception of Basic Tastes and Astringency Relate with Dietary Intake and Preferences for Fruits and Vegetables

**DOI:** 10.3390/foods10081961

**Published:** 2021-08-23

**Authors:** Teresa Louro, Carla Simões, Paula Midori Castelo, Fernando Capela e Silva, Henrique Luis, Pedro Moreira, Elsa Lamy

**Affiliations:** 1Mediterranean Institute for Agriculture, Environment and Development (MED), Institute for Advanced Studies and Research, IIFA University of Évora, 7006-554 Évora, Portugal; teresalouro@hotmail.com (T.L.); carlasimoes3@hotmail.com (C.S.); fcs@uevora.pt (F.C.e.S.); 2Department of Pharmaceutical Sciences, Universidade Federal de São Paulo (UNIFESP), Diadema 09972-270, Brazil; pcastelo@yahoo.com; 3Department of Medical and Health Sciences, School of Health and Human Development, University of Évora, 7006-554 Évora, Portugal; 4School of Dental Medicine, University of Lisbon, 1649-003 Lisbon, Portugal; henrique.luis@fmd.ulisboa.pt; 5ciTechCare, Polytechnic of Leiria, 2410-541 Leiria, Portugal; 6Faculty of Nutrition and Food Sciences, University of Porto, 4200-465 Porto, Portugal; pedromoreira@fcna.up.pt; 7EPIUnit—Institute of Public Health, University of Porto, 4200-465 Porto, Portugal; 8Research Centre in Physical Activity, Health and Leisure, University of Porto, 4200-465 Porto, Portugal

**Keywords:** astringency, basic tastes, dietary intake, oral food perception, fruits, vegetables

## Abstract

*Background*: Oral food perception plays a major role in food acceptance, although the way it relates with food preferences and final choices in adults is still debatable. The objective of the present study was to assess the relationship between gustatory function, dietary habits and fruit and vegetable preferences. *Methods*: Recognition thresholds, suprathreshold and hedonics were accessed for sweet, bitter, sour, salty and astringency in 291 adult participants. A Food Frequency Questionnaire (FFQ) and a questionnaire for assessment of preferences for individual fruit and vegetables were filled by the participants. *Results*: Three clusters were obtained: “most sensitive”, “less sensitive” and “less sensitive only for sour”. The less sensitive cluster showed lower preferences for fruit and vegetables and higher intake of sweets and fast foods, whereas higher preferences for sweet veggies were observed in the “most sensitive” cluster. Basic tastes and astringency hedonics did not associate with fruit and vegetable preferences, but the sensitivity for these oral sensations did. *Conclusions*: Taste and astringency sensitivities are related with the preference for fruit and vegetables, being also associated with some dietary habits. The effectiveness of the strategies to promote plant-based healthy food consumption may benefit from the knowledge of individuals’ gustatory function.

## 1. Introduction

Oral sensory perception is essential in food choices, being recognized as a major driver of food preferences. From an evolutionary perspective, taste had an essential role in species survival, informing about the presence of nutrients or toxic compounds in foods [1]. However, individuals vary among each other in the intensity with which they perceive the different sensory characteristics of foods and this may affect their food acceptance and, consequently, their final choices [2]. Since it is known that dietary habits are linked to the development of chronic diseases (e.g., obesity, diabetes, hypertension, among others) and quality of life, oral sensory perception is a particularly important aspect to consider when developing strategies to prevent these diseases and to promote healthy eating. Moreover, the understanding on the relationship between oral sensory perception and food preferences is particularly important for food industry, for the development and formulation of products accepted by consumers, as well as for agricultural and environmental actors, for promoting sustainable food consumption.

Taste and trigeminal oral sensations can be considered in terms of quality, intensity, and pleasantness/unpleasantness. Taste acuity can be measured at threshold and supra-threshold level. Threshold level, either detection or recognition, means the minimum concentration of a stimulus that is either detected or recognized, respectively. Concerning suprathreshold level, this refers to the intensity (magnitude) of a stimulus perceived at a concentration above the threshold. In oral sensory perception we may also consider the affective value associated to that sensation; as this is not an objective method for evaluating taste function, it provides important information that links stimulus reception ability to the subjective feelings that the stimuli elicit [3,4]. Considering these different dimensions of oral sensory perception, their relationship with food intake, accessed by different authors, results in some controversy about the role of oral perception in the final choices and dietary habits. 

Presently, five basic tastes are recognized: sweet, umami, salty, sour and bitter. Each of these tastes are associated with innate behavior: sweet, salty, and umami tastes are innately accepted and trigger ingestion, while in contrast bitter and sour are associated with food aversion [5]. Differences in bitter taste sensitivity have been linked to vegetable acceptance and consumption. In the case of the bitter compound 6-*n*-propylthiouracil (PROP), a link between the sensitivity and *Brassica* intake was reported [6]. However, this relationship was not observed in all studies; for example, a recent meta-analysis reported no clear relationship between taste perception and food choices or intake in adolescents [7]. Like bitterness, sweet taste perception has also been suggested to be related with food intake. Besides sweetness being a potential factor influencing the consumption of sweet foods, its association with the preference of bitter/sour vegetables was also observed [2]. Tan and Tucker [8] reviewed the link between sweetness perception and intake and found only a small proportion of studies reporting associations between sweetness sensitivity or intensity with energy (or carbohydrate or sugar) intake. In cases where an association was observed, higher sensitivity or intensity perception of sweetness appeared negatively associated with intake. For hedonics, the results about a relationship between sweetness preference and energy intake were more consistent and, in this case, the association was positive. 

Salt is a taste linked to the presence of sodium in food. From an ecological perspective, a moderate level of saltiness is preferred, over absence, whereas increased levels are rejected, all to ensure an adequate level of salt. However, a considerable high variability of reactions to salty taste exists and this appears to be, at least partially, due to the high influence that learning has on salt preference [9]. The initial use of salt for preservative purposes, in many societies, can be responsible for increased levels of sodium in foods and the consequent current high consumption [10]. Sourness, to some extent, is linked to innate rejection in newborns, although not as strong as bitterness rejection [11]. A direct influence of sourness preference in food consumption was demonstrated in studies where infants and children with higher preferences for citric acid were observed to consume more fruit than children less tolerant to this taste [12]. Umami taste appears to be a taste indicating the presence of protein, since it is positively correlated with this macronutrient [13]. A study from Kubota and colleagues [14], in Asian individuals, observed different preference levels for foods like shellfish, tomato, carrot, milk, low fat milk, cheese, dried shiitake, and kombu, among individuals with different umami sensitivities, while no association was observed between umami sensitivity and total energy consumption. In an European population, the association of umami taste and food intake was also observed, being this positive for non-cruciferous vegetables [15]. 

Besides basic tastes, astringency is also an important sensory characteristic, which is present in several vegetable-based foods. Astringency is a tactile sensation felt as a dry, rough feeling in the mouth, which is usually due to the interaction of polyphenols (mainly tannins) with salivary proteins. However, this sensation can be also induced by alums, acids and some metals. Although the participation of salivary proteins in astringency development is accepted, the interaction of astringent compounds with oral cells has been also considered, and a definitive theory is still lacking due to the complex oral sensations [16]. The level of intensity with which astringency is perceived varies among individuals and appears to affect the acceptance of polyphenol rich foods [17]. Due to the role that polyphenols have as antioxidants, the ingestion of foods rich in these compounds is particularly important to health. Despite of this, the contribution that the sensitivity to astringency or even the acceptance level (hedonics) has in food habits has been few explored. 

The objective of the present study is to assess how individual variations in the perception of basic tastes and astringency relates with dietary habits and with preferences for fruit and vegetables. For this, three dimensions of perception (thresholds, intensities and hedonics) were considered. Our hypothesis is that gustatory function and astringency are associated with food/nutrients intake frequency and with fruit and/or vegetables preferences.

## 2. Materials and Methods

### 2.1. Participants

The participants were recruited from public institutions, such as municipalities and public schools. In total, 291, 20–59 year-old adult participants (146 men) from the Alentejo region of Portugal participated in the study. In order to include a representative sample of the region, a sample was constituted based on the number of men and women from each age group that represent the percentages reported in the Official National Database Contemporary Portugal (PORDATA; https://www.pordata.pt/Home; accessed on 15 March 2016) for those groups. The exclusion criteria included pregnancy and presence of acute or chronic respiratory diseases that can affect food sensory perception. Smoking was not an exclusion criterion. Written informed consent was signed by all of the subjects and it has been performed in accordance with the Declaration of Helsinki.

### 2.2. Anthropometric Evaluation

Height and weight were accessed using a portable stadiometer (mod. 803, Seca, Hamburg, Germany) and a digital scale accurate to the nearest 0.1 Kg (Seca 803). Measurements were performed according to the European Health Examination Survey procedures. Participants were evaluated in a stand position wearing light cloths and barefoot. Body mass index (BMI, Kg/m^2^) was used to classify individuals as underweight (BMI < 18.5), normal-weight (18.5 ≤ BMI < 25), overweight (25 ≤ BMI < 30) and obese (BMI ≥ 30), respectively [18].

### 2.3. Dietary Intake Assessment

The frequency of food consumption, in the previous 12 months, was assessed using a self-administered, semi-quantitative food frequency questionnaire (FFQ), validated for Portuguese adults [19], which included 86-food items. Before questionnaire completion, participants were instructed about how questionnaires should be filled. Food intake was calculated by multiplying one of the nine possibilities of frequency of consumption (from—never or less than once per month, to six or more times a day), by the weight of the standard portion size of the food-item. Energy and nutritional intake were estimated using an adapted Portuguese version of the nutritional analysis software Food Processor Plus (ESHA Research Inc., Salem, OR, USA).

### 2.4. Preference for Fruit and Vegetables

To assess potential relationship between taste perception and the preference for fruit and vegetables, a second questionnaire was filled, which assessed the frequency of consumption of individual vegetables and fruit. These two food groups were chosen, since these are foods which consumption is important for health and that can be eaten in raw-form or with minimal processing. In this questionnaire, the same frequency response options that were previously used in FFQ were considered: “3 or more times a day”, “2 times per day”, “1 time per day”, “5–6 days per week”, “3–4 days per week,” “1–2 days per week”, “2–3 times per month”, “1 time per month”, “never or less than 1 time per month”. Additionally, the preference level, for each of these items, was assessed using a 5-point scale. Fruit and vegetable questionnaire is presented in Appendix A. For analysis purposes, groups of vegetables and fruits with characteristic taste profile were formed, using the classification from van Langeveld and colleagues [20]. These groups were: (1) sweet vegetables—carrots and beans; (2) sweet-sour vegetables—tomato; (3) neutral vegetables—broccoli, cabbages, courgette, cauliflower, leek, mushrooms, cucumber, spinach; (4) bitter vegetables—brussels sprouts, lettuce, soya beans, onion; sweet fruits—banana, pear; sweet-sour fruits—grape, pineapple; sour fruits—apple, kiwi, orange, tangerine, strawberry.

### 2.5. Taste Sensitivity

Taste perception was evaluated using filter paper taste strips prepared in our laboratory, as previously described and validated for the Portuguese population [21]. Four concentrations of solutions of four basic tastes (bitter, sweet, salty, and sour) were used for preparing the taste strips (Table 1). The solutions were prepared with distilled water using the concentrations reported in Table 1. The part of the filter paper strip that comes into contact with the tongue was cut into a circle, 2 cm in diameter. The filter papers were soaked in the solutions, and oven dried at 60 °C. Control strips were prepared in the same way, by soaking the strip in distilled water, instead of taste solutions. In the case of astringency, due to the specific characteristics of this sensation, namely the tactile aspects involved, filter paper taste strips were observed to be not adequate. As such, four solutions of different concentrations of tannic acid were chosen, based on previous works evaluating inter-individual variability in this oral sensation [17] (Table 1). Three types of answers were requested by each participant: (1) if he/she perceived any taste different from the control taste strip (made using distilled water) or distilled water (in the case of astringency tests), for each of taste strip/solution and was able to recognize it; (2) for the highest concentration of each stimulus, an intensity judgment on a labeled magnitude scale (LMS) (ranging from “barely detectable” to “strongest imaginable sensation”), as previously described [22]. In the case of astringency, this intensity assessment was done for the second concentration, instead of the fourth; (3) for the concentration tested for intensity, hedonics was also assessed, by requesting individuals to choose the preference level in a 5-point scale (being 1 “hated” and 5 “highly preferred”).

The tests were performed in the institutions/locations that agreed to participate. The sensory tests were all performed in a quiet room with adequate light, temperature and humidity conditions, ensuring that all participants could be calm and concentrated while performing the requested tasks. Some days before the test session day, the subjects received a detailed document, with the explanation of the experiment and instructions. All participants were instructed to refrain from the intake of food and beverages other than water, chewing gum, and smoking for at least 1 h prior to testing. All tests were performed during the morning, between 10 h and 11 h, with the participation of one researcher, who guided the participants.

### 2.6. Statistical Analysis

Statistical analysis was performed using AMOS 25 (IBM) and SPSS 27.0 software (IBM). Descriptive statistics consisted of means, standard deviation, medians, quartiles, and percentages. Normality and homocedasticity was assessed using Shapiro-Wilk and Levene test, respectively. Non-parametric Mann-Whitney test was used to compare continuous variables between sexes, whereas Chi-square test was used for categorical variables comparison. The results for continuous variables were presented as means and standard deviation or medians and interquartile range, whereas categorical variables were presented as percentages. Taste function parameters were correlated using the non-parametric Kendall’s-tau test.

In order to reduce the number of variables analyzed using the FFQ, principal component analysis was used to estimate the number of latent variables emerging from the 86-food items. For this purpose, the correlation matrix was examined, and the decision of the number of components to be retained was based on the total of explained variance and Scree plot examination. The Varimax rotation was performed and the overall Kaiser-Meyer-Olkin (KMO) measure and Bartlett’s test of sphericity were examined as assumptions of the test.

K-means cluster analysis was performed to identify groups of participants with similar taste sensitivity, dietary habits and preferences. The analysis included the following variables: age, sex, BMI, taste/astringency thresholds and taste/astringency preferences, preferences for sweet, sour, bitter, and neutral fruit and vegetables, and the component loadings generated from principal component analysis (the ones retained, as described above). The final number of clusters was based on the interpretability and reliability of the cluster solution; the silhouette coefficient and the differences between clusters assessed by One-way ANOVA test were used for clustering validation.

Finally, to understand the relationship between oral sensory perception and fruit and vegetables preferences, a structural equation model (SEM) was used combining measured variables and latent constructs. The measurement model (path analysis) was defined *a priori* and was built by the maximum likelihood method and considered the following assumptions: χ2 test, degrees of freedom, root mean square error of approximation (RMSEA), Confirmatory Fit (CFI) and Tucker-Lewis Index (TLI) fit indices.

## 3. Results

### 3.1. Participant’s Characteristics

The participants’ characteristics are presented in Table 2. Men and women did not differ in BMI or age but presented differences in sweet and salty taste recognition thresholds, being these lower in women than in men. Comparing intensities, this higher sensitivity was also observed in women, but only in the case of salty taste.

### 3.2. Relationship between Taste Sensitivity and Taste Preferences

Although the correlations are weak, it was observed that the individuals with higher sensitivity for one basic taste tend to be more sensitive to the other basic tastes (Table 3). When the correlation between taste sensitivity and taste preferences was tested, it was observed a negative correlation between sweet taste threshold and the preference for this taste (R = −0.131; *p* = 0.024; *n* = 245), i.e., a tendency of most sensitive individuals to have higher preference for sweet taste. For bitter taste, the opposite occurs: higher thresholds (lower sensitivity) are associated with higher preference levels (R = 0.287; *p* = 0.0005; *n* = 245). For sour and salty tastes, no statistically significant correlation was observed between thresholds and hedonic ratings.

### 3.3. Relationship between Taste Function and Dietary Habits

In order to summarize the information gathered by FFQ, 86-food items were reduced into latent variables. Principal component analysis was run using Varimax rotation to facilitate the interpretation of results. The KMO measure was 0.60 and the results of the Bartlett sphericity test was significant (*p* < 0.001). By observing the inflection point in the scree plot, 15 components were retained that collectively explained 50% of the total variance (Table 4). Moreover, these components resulted in meaningful food combinations. Based on Table 4, it is possible to see that the FFQ food items were distributed into the following components: Component 1 (high variety foods); Component 2 (white meat, sea food and not-salad vegetables); Component 3 (sausages); Component 4 (sweets); Component 5 (fats); Component 6 (fast-food); Component 7 (alcoholic beverages and coke); Component 8 (animal protein); Component 9 (bread, half-fat milk and jam); Component 10 (complex carbohydrate); Component 11 (low-fat fish, pulses and broccoli); Component 12 (sea food and salad); Component 13 (low fat milk and yoghurt); Component 14 (cheese, nuts, olives and salad); and Component 15 (“common” fruits).

Further, K-means cluster analysis was run to identify groups of similar individuals based on oral sensory sensitivities and preferences, fruit and vegetables preferences and dietary habits. The analysis included the variables: age, sex, BMI, taste thresholds and preferences for sweet, salt, sour, bitter and astringency, the preferences for sweet-sour veggies, bitter veggies, neutral veggies, sweet veggies, sweet-sour fruits, sour fruits and sweet fruits, and the fifteen components generated from QFA. The analysis identified three reliable and meaningful clusters. Table 5 shows the taxonomy description of the three clusters: Cluster 1 (*n* = 81) was characterized by young participants with higher preferences for sweet and salty tastes; they also showed lower preferences for fruit and vegetables and higher intake of sweets, fast-food, animal protein, complex carbohydrates, and “common” fruits (apple, pear, banana, orange and kiwi). Cluster 2 (*n* = 100) was composed of individuals with intermediate age, with lower taste thresholds (higher sensitivity) and higher preferences for fruit and vegetables; a higher intake of low-fat milk and yoghurts together with lower intake of complex carbohydrates and sausages was also observed. Finally, Cluster 3 (*n* = 100) was composed of middle-aged individuals (mean age = 53 years) with higher BMI (76% were overweight), higher threshold for sour taste, higher preference for sweet and sour vegetables, and higher intake of bread, half-fat milk, cheese, salad, nuts and olives. On the other hand, the intake of sweets, fast-food, fats and animal protein was lower in Cluster 3 when compared to the other ones. 

### 3.4. Relationship between Taste Function and Fruit & Vegetables Preference

In order to understand the relationship between taste threshold, taste preference, and fruit and vegetables preferences, a structural equation model was defined a priori (Figure 1) and built by the maximum likelihood method. The model showed a good fit, and the following parameters were found: Chi-square Mean/Degree of Freedom (CMIN/DF) = 2.435, CFI = 0.822, TLI = 0.761, and RMSEA = 0.072.

According to the analysis, it was possible to observe that all coefficients for taste thresholds and preferences were significant (*p* < 0.05), except for sweet preference (*p* = 0.270). Additionally, all observed variables for fruit and vegetables preferences showed significant coefficients. The model was specified by the following expression:Fruit & vegetable preferences = −0.22 × taste threshold − 0.02 × taste preference

Taste threshold significantly influences the preferences for fruit and vegetables (*p* = 0.014), while the influence of taste preference was not significant (*p* = 0.879). Moreover, taste threshold did not significantly influence the taste preference (*p* = 0.362) (Figure 1).

## 4. Discussion

The present study investigated the association of oral perception, namely basic tastes and astringency, with dietary habits and preferences for fruit and vegetables in adults. To our knowledge, there are only a few studies testing this relationship (e.g., [14,15,23]) and the existing ones did not evaluate astringency. Considering the importance that sensory aspects have for food acceptance, choices and intake, an adequate understanding of the influence that inter-individual differences in sensory perception have in long-term habits can be of major importance to ensure good nutritional management and to define strategies to strengthen healthy eating behaviors.

Taste and astringency perceptions were assessed by three different means: (i) recognition thresholds; (ii) intensity ratings of a recognized concentration; and (iii) hedonic ratings. Recognition thresholds for the different tastes correlated positively. The same was observed for intensity ratings, but when astringency was considered, a lack of correlation existed for saltiness and astringency intensity ratings. This positive correlation between different basic tastes perception was also shown in previous studies (e.g., [3,24]). However, all the correlations, despite statistically significant, were weak to moderate, suggesting that individual tastes have a certain level of independency from each other and are subjected to other influences.

Concerning the association between sensory perception and preference, the present results showed a positive correlation between sweet taste sensitivity and sweet hedonic ratings, in line with a recent study [25]. On the other hand, for astringency and the other three basic tastes evaluated (bitter, sour and salty), higher sensitivities are associated with lower hedonics. High levels of saltiness and sourness are known to be unpleasant [26], as well as bitterness and astringency, for which increases in intensity perception have been seen associated to lower acceptance [17]. Bitterness and astringency levels are associated with the potential presence of toxic and/or non-nutritional compounds and induce innate aversion [1].

In line with previous reports, it was observed that men and women differ in their taste sensitivities, with women being more sensitive to sweet and salty than men. Some studies, comparing taste function between sexes, also reported higher taste sensitivity in females, comparatively to males (e.g., [27,28,29]), although some others only found differences between some age groups [30]. Different methodologies for assessing taste function and evaluations in populations with different genotypes, food habits, and health conditions may explain these diverse results among studies. 

The relationship between obesity and taste sensitivity is not a consensual issue. Some studies had suggested that individuals with obesity have lower taste acuity than normal-weight pairs (e.g., [2,31,32,33]), although others observed higher sensitivity to salty and sweet taste in participants with obesity [34] and others suggested no relationship at all [35,36]. In the present study, individuals were grouped into three clusters, from which it was possible to observe the individuals less sensitive to sour having higher BMI. Whereas a considerable number of studies reported associations between sweetness and bitterness with BMI, only few reported decreased sourness sensitivity in individuals with obesity [37,38]. The logical explanation for this lower sour taste sensitivity in high BMI individuals is difficult to provide considering the data collected in this study. However, it is interesting to note that these individuals are also low sensitive to bitter taste. This negative association between BMI and bitterness contradicts previous results obtained in children, in which higher sensitivity to bitterness predicted obesity [2], but are in accordance with studies in adults [33]. Once more, different techniques to assess taste sensitivity, and differences in the populations studied (children vs. adults) are all possibilities to explain the different results found.

The main aim of this study was to assess if oral taste and astringency perception is associated with dietary habits and fruit and vegetables preference. The classification of individuals into three clusters (individuals less sensitive to most tastes, individuals highly sensitive to most tastes, or individuals with low sensitivity to sour taste) showed an association between taste function and dietary habits. Astringency hedonics did not differ between clusters, what may be explained by being a sensation rejected by almost individuals, particularly when not in food matrix. Individuals that are less sensitive to sweet, bitter, and salty, but that, at the same time, have higher preferences for sweetness and saltiness, are the ones with higher intake of sweets, fast-food and foods of animal origin, i.e., unhealthy foods. This cluster is composed by younger adults, in line with the results of other authors, who also observed poorer nutritional habits in young, compared to older adults [39]. It remains to be clarified if the lower sweet and salt tastes sensitivities are the causes or consequences of the higher intake of these food types. 

On the opposite, the cluster with individuals with higher sensitivity for sweet, bitter and salty tastes is the one with higher preferences for fruit and vegetables and higher intake of low-fat dairy and salads. Despite it would be expected that a higher sensitivity to bitterness could be associated with lower preference for bitter foods, such as vegetables, it is possible to hypothesize that the simultaneous higher sensitivity to sweetness can allow individuals to perceive some sweetness from the complex food matrix of vegetables, let them take higher pleasure from these foods. This hypothesis is supported by previous observations in children, in which girls with higher sweet taste sensitivity had higher hedonic ratings of sour-bitter vegetables [2]. Different studies related vegetable intake with oral perception, particularly with bitter taste [40] and astringency function [17] and, whereas some of the studies report lower intake of vegetables by bitter taste [40] or astringent [17] sensitive individuals, others fail to find such a relationship [41]. Vegetables are eaten cooked, in opposition to fruits, and different cooking methods may explain the difficulty in observing consistent relationships.

The cluster constituted by individuals less sensitive to sour is composed mainly by individuals with overweight. Whereas the intake of fast-food and sweets is reduced in this group, comparatively to the other two clusters, the intake of sausages and alcoholic beverages was high, with the potential contribution of these to the increased BMI. Nevertheless, it is possible that some of the habits reported for these individuals are underestimated, particularly at the level of fats and sweets, since individuals with excess weight tend to underreport consumption [42].

One particularly interesting result is that fruit and vegetable preference showed a stronger association with taste sensitivity than with taste preference. Through the results obtained with the Structural Equation Model, it is possible to see that sweet and bitter taste sensitivities have the higher contribution to the latent variable “taste sensitivity” and this influences particularly fruit preferences. Fresh fruits are consumed without cooking/processing, and this may explain a higher influence of taste sensitivity in fruit preference than in vegetables, for which a diversity of cooking procedures can be considered, masking the original taste, as stated above.

Some limitations need to be considered in the present study. Due to the cross-sectional study design, the direction of the association between sensory functions and parameters such as BMI or dietary behavior remains unclear. Since data was collected at one time point, it is not possible to say whether it is the sensory acuity that results in food choices (which define dietary habits), or whether it is eating habits that shape sensory perception and preference.

Umami and fat (taste) was not accessed in this study. These are sensory characteristics potentially associated with the intake of protein and fat-rich foods. As such, some influence of the sensitivity/preference for these stimuli in some of the associations (or lack of associations) between gustatory function and dietary habits cannot be excluded.

In this study, factors like smoking habits or chronic medication were not considered. Taking into account that they can affect taste function, further studies including these potential confounding factors are of interest.

## Figures and Tables

**Figure 1 foods-10-01961-f001:**
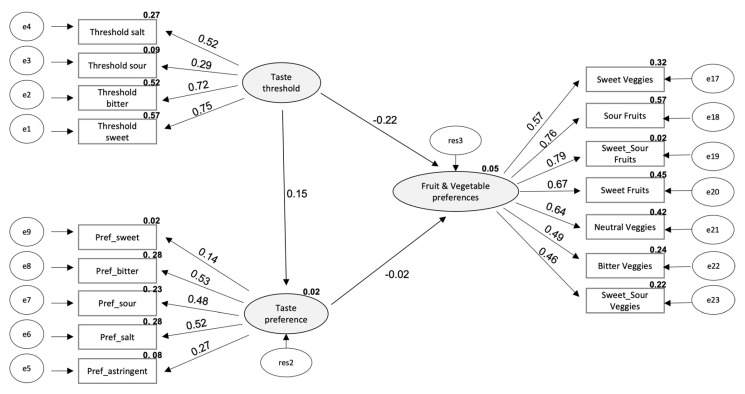
Structural equation model defined *a priori* to observe the relationship between taste threshold, taste preference, and fruit and vegetables preferences (Fruit & Vegetable preferences = −0.22 × taste threshold −0.02 × taste preference). Ellipses indicates latent unobservable constructs; box indicate observed variable; arrows indicate direct effects (e = error; res = residuals).

**Table 1 foods-10-01961-t001:** Concentrations (g/mL) of solutions of four basic tastes and astringency.

Oral Sensation	Compound	Conc 1	Conc 2	Conc 3	Conc 4
Bitter	Quinine hydrochloride	0.0004	0.0009	0.0024	0.0060
Sweet	Sucrose	0.05	0.10	0.20	0.40
Sour	Citric acid	0.050	0.090	0.165	0.300
Salty	Sodium chloride (NaCl)	0.016	0.040	0.100	0.250
Astringent	Tannic acid	0.0017	0.0023	0.0032	0.0045

**Table 2 foods-10-01961-t002:** Participants’ characteristics and anthropometric data.

		Men (*n* = 132)	Women (*n* = 137)	*p*-Value
Age—median [range]		41.5 [33.0–52.0]	42.0 [33.5–51.0]	>0.05
BMI—median [range]		27.0 [24.1–30.4]	25.9 [23.0–29.7]	>0.05
Urban		60	51	>0.05
Rural		72	86	
Sweet taste thresholds (g/mL)	0.050	70	79	0.0270 *
	0.100	28	43	
	0.200	18	7	
	»0.400	11	7	
Bitter taste thresholds (g/mL)	0.0004	64	79	0.501
	0.0009	23	21	
	0.0024	9	11	
	»0.0060	25	19	
Sour taste thresholds (g/mL)	0.050	97	97	0.625
	0.090	13	21	
	0.165	6	6	
	»0.300	9	8	
Salty taste thresholds (g/mg)	0.016	55	83	0.004 *
	0.040	28	26	
	0.100	13	14	
	»0.250	19	5	
Sweetness intensity—Median [range]		2.40 [1.40–3.83]	2.70 [1.40–4.10]	0.305
Bitterness intensity—Median [range]		5.15 [2.98–6.70]	5.30 [2.90–7.05]	0.962
Sourness intensity—Median [range]		5.95 [3.90–9.20]	5.80 [4.38–7.88]	0.526
Saltiness intensity—Median [range]		3.50 [0.08–5.20]	4.30 [2.58–5.90]	0.006 *
Astringency intensity—Median [range]		4.00 [2.23–5.80]	4.00 [3.18–5.53]	0.534
Sweetness preference (% within sex)	1	0.9	4.30	0.114
	2	5.7	11.1	
	3	14.2	17.1	
	4	67.9	62.4	
	5	11.3	5.1	
Bitterness preference (% within sex)	1	57.9	60.7	0.439
	2	32.5	34.4	
	3	7.9	3.3	
	4	0.9	1.6	
	5	0.9	0.0	
Sourness preference (% within sex)	1	53.9	60.0	0.716
	2	33.0	25.6	
	3	3.5	2.4	
	4	7.8	9.6	
	5	1.7	2.4	
Saltiness preference (% within sex)	1	20.7	26.8	0.142
	2	49.1	35.8	
	3	16.4	17.9	
	4	11.2	18.7	
	5	2.6	0.8	
Astringency preference (% within sex)	1	54.5	56.3	0.299
	2	34.5	33.6	
	3	10.9	6.7	
	4	0.0	2.5	
	5	0.0	0.8	

* Statistically significant at *p* < 0.05; » means higher or equal to.

**Table 3 foods-10-01961-t003:** Correlation between sensitivities for each of the basic tastes evaluated, either by detection thresholds or perception intensity.

		Thresholds			Intensities		
		Bitter	Sour	Salty	Bitter	Sour	Salty
**Thresholds**	Sweet	0.274 (*p* = 0.0005)	0.173 (*p* = 0.002)	0.205 (*p* = 0.0005)			
	Bitter		0.253 (*p* = 0.0005)	0.215 (*p* = 0.0005)			
	Sour			0.183 (*p* = 0.001)			
**Intensities**	Sweet				0.190 (*p* = 0.0005)	0.214 (*p* = 0.0005)	0.174 (*p* = 0.0005)
	Bitter					0.230 (*p* = 0.0005)	0.289 (*p* = 0.0005)
	Sour						0.278 (*p* = 0.0005)

**Table 4 foods-10-01961-t004:** Component loadings of food patterns obtained by principal component analysis with Varimax rotation.

Component	1	2	3	4	5	6	7	8	9	10	11	12	13	14	15
*% Cumulative variance*	7	13	18	21	25	27	30	33	35	38	40	42	45	47	50
Fat milk													−0.341		
Half-fat milk									0.318						
Low-fat milk													0.413		
Yoghurt													0.576		
Cheese														0.332	
Milk desert				0.550											
Ice cream											0.305				
Eggs								0.615							
Chicken								0.585							
Turkey or rabbit								0.605							
Cow or pig								0.448							
Liver															
Tongue			0.855												
Ham							0.404								
Sausage			0.752												
Bacon			0.631												
Fat fish								0.331							
Low-fat fish											0.830				
Cod								0.429							
Canned fish						0.320		0.400							
Squid or octopus		0.318								0.374		0.337			
Shrimp		0.315										0.493			
Olive oil					0.670										
Vegetable oil					0.718										
Margarine					0.721										
Butter					0.603										
White bread									0.419				−0.398		
Whole grain bread															
Corn bread												0.321			
Breakfast cereals										0.427					
Rice										0.826					
Pasta										0.754					
Fried potato (home made)													−0.317		
Fried potato (industrial)													−0.412		
Boiled potato										0.544					
Low-sugar cookies					0.389		−0.322								
Cookies				0.392					0.322						
Pastry				0.543											
Chocolate				0.608					0.368						
Chocolate snacks				0.666											
Jam									0.481						
Sugar				0.351										0.325	
White cucumber		0.643													
“Penca” cucumber		0.634	0.368												
“Galega” cucumber		0.732													
Broccoli		0.537									0.601				
Cauliflower		0.663													
Cabbage sprouts		0.718													
Green bean		0.705													
Lettuce												0.317		0.478	
Onion												0.423		0.469	
Carrot												0.355	0.301	0.369	
Turnip		0.477											0.360		
Tomato												0.664			
Pepper			0.665									0.347			
Cucumber												0.550			
Pulses											0.750				
Peas		0.432													
Apple/pear									0.333						0.593
Orange	0.428														0.401
Banana															0.683
Kiwi									−0.315						0.529
Strawberry	0.813														
Cherries	0.747														
Peaches	0.773														
Melon	0.793														
Persimmon	0.485								0.453						
Fig	0.809														
Grape	0.634														
Canned fruit				0.424											
Nuts														0.470	
Olives														0.577	
Wine							0.445								
Beer							0.685								
Spirits							0.556								
Sweetened beverages							0.538								
Ice Tea															0.348
Coke							0.517								
Coffee															
Black tea								0.376							
Processed															
Mayonnaise						0.675									
Ketchup						0.648									
Pizza						0.652									
Hamburger						0.308					0.383				
Vegetable Soup									0.450						

Coefficients less than 0.30 are omitted.

**Table 5 foods-10-01961-t005:** Final cluster centers (means) of the taste and nutritional variables (important differences which identify the clusters are shown in light gray color).

	Cluster			
	1	2	3	Z
	“Less Sensitive”	“Most Sensitive”	“Less Sensitive to Sour”	
	(*n* = 81)	(*n* = 100)	(*n* = 100)	
Sex	50% women	54% women	49% women	0.25
Age	28	41	53	1045.96
BMI	26.08	26.36	28.21	6.25
Threshold sweet	0.15	0.08	0.14	3.23
Pref sweet	4	4	3	3.42
Threshold bitter	0.0067	0.0037	0.0064	1.23
Pref bitter	2	1	2	1.21
Threshold sour	0.09	0.07	0.12	3.49
Pref sour	2	2	2	2.51
Threshold salty	0.084	0.050	0.079	1.91
Pref salty	3	2	2	5.73
Pref Astringency	2	2	2	0.35
Pref Sweet_Sour Veggies	3.92	4.18	4.20	2.05
Pref Bitter Veggies	3.55	3.80	3.63	3.19
Pref Neutral Veggies	3.60	3.91	3.89	6.97
Pref Sweet Veggies	3.82	4.27	4.09	9.18
Pref Sweet_Sour Fruits	3.96	4.29	4.23	6.43
Pref Sour Fruits	4.07	4.24	4.20	2.32
Pref Sweet Fruits	4.04	4.28	4.25	4.38
Comp 1 (high variety fruits)	−0.00622	−0.07148	0.07580	0.54
Comp 2 (white meat, sea food and brassica)	−0.13108	0.06519	0.04164	0.99
Comp 3 (sausages)	−0.03615	−0.10758	0.13578	1.55
Comp 4 (sweets)	0.26724	−0.00134	−0.21513	5.37
Comp 5 (fats)	0.02783	0.06891	−0.09077	0.68
Comp 6 (fast-food)	0.33285	0.04326	−0.31243	10.08
Comp 7 (alcoholic beverages and coke)	−0.00182	−0.07415	0.07488	0.55
Comp 8 (animal protein)	0.13062	0.04130	−0.14669	1.86
Comp 9 (bread, half-fat milk)	−0.29482	0.03059	0.20851	5.94
Comp 10 (complex carbohydrates)	0.28476	−0.15167	−0.08050	4.88
Comp 11 (low-fat fish, pulses and broccoli)	0.05189	0.04580	−0.08738	0.59
Comp 12 (sea food and salad)	−0.03061	0.05494	−0.02959	0.23
Comp 13 (low-fat milk and yoghurt)	0.08889	0.12334	−0.19411	3.0
Comp 14 (cheese, nuts, olives and salad)	−0.04713	−0.13186	0.16872	2.40
Comp 15 (“common” fruits)	0.14243	−0.13694	0.02020	1.78

BMI, body mass index; pref, preference; comp, component.

## Data Availability

The data presented in this study are available on request from the corresponding author. All data relevant to the study are included in the article and access to raw data would be provided upon request.

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
