# Peer review of "How Individual Variations in the Perception of Basic Tastes and Astringency Relate with Dietary Intake and Preferences for Fruits and Vegetables"

_foods, 2021, doi:10.3390/foods10081961_

Round 1

Reviewer 1 Report

COMMENTS AND SUGGESTIONS FOR AUTHORS

The paper by Louro et al “How individual variations in the perception of basic tastes and astringency relates with dietary intake and preferences for fruits and vegetables” describes the results of a study on the relationship between gustatory function, dietary habits and fruits and vegetable preferences. The authors stated that the objective of the study was to assess how individual variations in the perception of basic tastes and astringency relates with dietary habits and with preferences for fruits and vegetables

Comments:

Majors

The authors enrolled in the study 291 adult volunteers, including smokers. However, previous studies suggest that smoking influences salt perception (10.14800/sp.1133, https://doi.org/10.1096/fasebj.30.1_supplement.lb403 ) and that subjects with lower salt taste threshold were more likely to consume more fruit (https://doi.org/10.3390/nu12041164). Nevertheless, the authors did not take smoking into account when setting up their analysis models. For this reason, if it is possible the analyses should be done again considering the smoke. If this is not possible, the work has serious flaws.

Minor

Discussion lines 329-30 :…From our knowledge, there are only few studies testing this relationship and the ones existing did not evaluate astringency….  These studies should be cited

Author Response

COMMENTS AND SUGGESTIONS FOR AUTHORS

The paper by Louro et al “How individual variations in the perception of basic tastes and astringency relates with dietary intake and preferences for fruits and vegetables” describes the results of a study on the relationship between gustatory function, dietary habits and fruits and vegetable preferences. The authors stated that the objective of the study was to assess how individual variations in the perception of basic tastes and astringency relates with dietary habits and with preferences for fruits and vegetables

Comments:

Majors

The authors enrolled in the study 291 adult volunteers, including smokers. However, previous studies suggest that smoking influences salt perception (10.14800/sp.1133, https://doi.org/10.1096/fasebj.30.1_supplement.lb403 ) and that subjects with lower salt taste threshold were more likely to consume more fruit (https://doi.org/10.3390/nu12041164). Nevertheless, the authors did not take smoking into account when setting up their analysis models. For this reason, if it is possible the analyses should be done again considering the smoke. If this is not possible, the work has serious flaws.

Answer – We understand the reviewer concern, since smoking can be one factor affecting gustatory function. We were aware of that when designing the study, but we decided to not consider smoking, as well as not considering other factors of gustatory function change (like medicine drugs, for example - https://www.ncbi.nlm.nih.gov/pmc/articles/PMC2980431/), since our main aim was to access how the way taste is perceived can be related with food preferences and food habits. Independent of the reasons for a particular taste capacity, we want to see how that could be related with the parameters referred. For that aim, we believe that not including smoking, medicines, etc., does not limit the conclusions. We excluded acute problems, that could affect gustatory function, because acute changes could lead to errors when relating parameters of medium term, like dietary habits. Smoking, however, is expected to be maintained for several years, or at least months.

Nevertheless, and agreeing with the reviewer that could be also interesting to have such information, we highlighted this limitation of the study, at the end of discussion section.

Minor

Discussion lines 329-30 :…From our knowledge, there are only few studies testing this relationship and the ones existing did not evaluate astringency….  These studies should be cited

Answer – Done

Reviewer 2 Report

This manuscript describes a substantial volume of research, with very clear and interesting results. It appears to be the first study to comprehensively assess both sensitivity and hedonic value (liking) for each of the basic tastes, together with diet, both reported intake and reported preferences. By including this wide range of data and systematically analysing the outcomes, the study succeeded to not only detect general correlations, but also quantify the relations among the different types of factors.

The study is worth replicating in other countries with different diets, and the clear description of the methodology will support such efforts.

This type of information can be very important to support the efforts to increase intake of vegetables and fruits, in particular if future similar studies confirm that the relations are general; although it would also be important f it is demonstrated that they depend on the food culture!

The English language is mostly good, but still needs careful proofreading. For example, the sentence in lines 23-25 is not clear, probably a language or editing problem; the sentences in lines 104-106 and 130-132 each contain several grammatical errors (although in those cases the meaning is clear). Numerous sentences throughout the manuscript contain single minor grammatical errors…

Use ‘overweight’ instead of ‘pre-obese’ (line 134).

Author Response

This manuscript describes a substantial volume of research, with very clear and interesting results. It appears to be the first study to comprehensively assess both sensitivity and hedonic value (liking) for each of the basic tastes, together with diet, both reported intake and reported preferences. By including this wide range of data and systematically analysing the outcomes, the study succeeded to not only detect general correlations, but also quantify the relations among the different types of factors.

The study is worth replicating in other countries with different diets, and the clear description of the methodology will support such efforts.

This type of information can be very important to support the efforts to increase intake of vegetables and fruits, in particular if future similar studies confirm that the relations are general; although it would also be important f it is demonstrated that they depend on the food culture!

 Answer – We would like to thank the reviewer for the so nice comments about our study. We agree that it would be very interesting to replicate this study in different countries and we will try to do it.

The English language is mostly good, but still needs careful proofreading. For example, the sentence in lines 23-25 is not clear, probably a language or editing problem; the sentences in lines 104-106 and 130-132 each contain several grammatical errors (although in those cases the meaning is clear). Numerous sentences throughout the manuscript contain single minor grammatical errors…

Answer – English was reviewed.

Use ‘overweight’ instead of ‘pre-obese’ (line 134).

Answer - Done

Round 2

Reviewer 1 Report

The authors have responded satisfactorily to my comments, therefore the

manuscript has been sufficiently improved to warrant publication in Food.